# Application of PET/Sepiolite Nanocomposite Trays to Improve Food Quality

**DOI:** 10.3390/foods10061188

**Published:** 2021-05-25

**Authors:** Teresa Fernández-Menéndez, David García-López, Antonio Argüelles, Ana Fernández, Jaime Viña

**Affiliations:** 1Department of Materials Science and Metallurgical Engineering, University of Oviedo, 33203 Gijón, Spain; tere_asturias@yahoo.es; 2SMRC Automotive Interiors Spain S.L.U., 47800 Medina de Rioseco, Spain; dgarci12@smrc-automotive.com; 3Department of Construction and Manufacturing Engineering, University of Oviedo, 33203 Gijón, Spain; antonio@uniovi.es; 4Klöckner Pentaplast, 33128 Vegafriosa, Spain; ana.fernandez@kpfilms.com

**Keywords:** PET, sepiolite, nanocomposites, MAP, microbiological quality, chicken

## Abstract

New PET and nanosepiolite materials are produced for its application in innovative packaging with better performance. In our previous work, we demonstrate that the use of different percentages of sepiolite modified with different organosilanes improved mechanical and barrier properties of PET. Nanocomposites permeability can decrease up to 30% compared to that of pure PET and the mechanical analyses show that, although PET nanocomposites are more brittle than virgin PET, they are also harder. In the present work, we are going to study the properties of this innovative packaging with real food analyzing mechanical properties related to the product transport together with permeability and microbiological characteristics. At the same time, it has been seen that it is possible to lighten trays, which is very important both industrially and environmentally. On the other hand, a good quality packaging for food needs to ensure that organoleptic and physico-chemical characteristics of the product inside are not modified due to migration of any of the packaging material to the food itself. Results obtained in this work also show lower count of aerobic mesophilic bacteria and *Enterobacteriaceae* (EB), reducing the incidence of food contaminations by microorganisms.

## 1. Introduction

Food packaging has many useful functions, such as food containment, marketing, protection, and preservation during the shelf life of a product. In order to accomplish all this, a food packaging material must have enough strength to overcome its filling process, transport, and customer handling. At the same time, it needs to have the appropriate barrier properties for certain applications, such as in modified atmosphere packaging (MAP), and, of course, it needs to keep migration of packaging components to food to a minimum, complying with all regulation regarding Food Contact Materials (FCM), such as Food and Drug Administration (FDA) in USA, or European Regulation (EU) No 10/2011 of 14 January 2011.

Due to its low weight and versatility, among other things, polymers have been one of the most important materials used in food packaging. Plastic packaging plays a key role in protecting food from exterior virus and microorganisms as well as helping extend shelf life of packed food. However, it is also very important to keep migration of plastic components into food to a minimum, since that could ruin the product packed. Migration tests are regulated by legislation; thus, the amount of migrants allowed in the food is specified by the global migration limits. At the same time, the legislation records the lists of permitted substances to be used in food contact materials, this is the “positive list”. Each of this substance has its migration limit specified, in order to avoid food toxicity. The analysis done for controlling the amount of each substance in the packaging are called specific migration tests.

Fresh products (poultry, fruits, vegetables, etc.) are a vehicle for the transmission of bacterial, parasitic, and viral pathogens capable of causing human illness [1]. Within food packaging applications, improving shelf life of packed poultry is a huge challenge for the industry. Fresh poultry meat is highly popular among consumers and, at the same time, it is highly perishable (rapid microbial growth) leading to high economic losses [2,3]. Its shelf life depends mainly on poultry handling and processing (in the initial number of microorganisms) [4] or on its storage conditions through all the food chain.

One of the most important food packaging systems is MAP, where the food is packed, together with a certain mixture of gases that will keep freshness of the food, enhancing preservation, and extending shelf life. MPA requires a tray or base container and a lid to seal the packaged content. There are many plastic materials that could be used for this application, being poly(ethylene terephthalate) (PET) one of the most widely used. Depending on the amount of barrier required for the application, sometimes it is needed to apply a multilayer film with a high barrier polymer in it. In the present work, the base material used for the tray will be PET as well as for the lid. Adding the nanoclay to the PET it is expected to obtain trays with improve mechanical and barrier properties. One of the most used clays when talking about nanocomposites is montmorillonite (MMT). However, the clay chosen for the work has been sepiolite, which is a magnesium silicate with the following formula: Mg_8_Si_12_O_30_(OH)_2_(H_2_O)_4_ 8H_2_O. It is a fibrous clay with nanometric dimensions that vary between 0.2 and 3 µm in length, 10–30 nm in width, and 5–10 nm in thickness, which gives the sepiolite a high aspect ratio of about 27. In addition, sepiolite has a surface area of about 300 m^2^ g^−1^, and an outer layer of silanol groups. All these characteristics make the sepiolite perfect for surface modification with organosilanes and other reagents on its surface. It has also shown better mechanical properties than MMT in previous works [5].

The migration analysis of the trays, as well as microbiological tests are done to prove the possibility of using this material for food packaging, complying with actual legislation.

## 2. Materials

PET pellets, kindly supplied by LINPAC Packaging (Pravia, Spain), were from Novapet S.A. (Zaragoza, Spain). The pellets’ intrinsic viscosity (IV) was 0.81 dL/g. PET-EVOH-PE laminated sheet is a PET sheet laminated with an EVOH-PE flexible film (its structure being EVA/PA/EVOH/PA/PE). The sealing top film of the tray is a film coated with aluminum oxide (AlOx) (BOPET), Mylar^®^ 850 from DuPont Teijin Films UK Limited, Middlesbrough, England. Mylar^®^ 850 is a co-extruded, one side amorphous, heat sealable polyester film, suitable for use in contact with food. Its oxygen transmission rate is 56 cm^3^/m^2^/day, at 23 °C and 60/70% RH, for a 30 micron film. Two types of sepiolite were supplied by Tolsa S.A. (Madrid, Spain), one modified with 2% of 3-metracyloxypropil trimetoxysilane (MEMO, CAS 2530-85-2) and the other one with 2% of 3-aminopropyltriethoxysilane (AMEO, CAS 919-30-2). Both organomodifiers are suitable for food packaging with restrictions regarding the amount of absorbed substance by kg of product packed, as stated in Regulation CE 975/2009 for MEMO (0.05 mg by kg of packed product) and in Directive 2007/19/CE for AMEO (between 0.05 mg to 3 mg by kg of product in the package). The nanocomposite masters were produced at Repol S.L. (Almazora Castellón, Spain) in an industrial polyamide low shear extruder.

## 3. Experimental Part

In order to produce nanocomposites at industrial level, it was necessary to do the first steps at laboratory scale, as shown in previous works [6]. The materials used in this paper, are those found to be the best ones in terms of processability and mechanical properties.

The first step of nanocomposite fabrication at industrial level was to produce the masterbatches.

The PET/nanosepiolite masters were produced at Repol S.L. facilities. Conditions in the production plant were optimized to minimize PET matrix degradation, reducing humidity, and decreasing extrusion shear on the nanocomposites. Two different masters with 10% sepiolite each were produced. In one master, the sepiolite was previously modified with MEMO, and the other one with AMEO in Tolsa S.A. Then, these masters were diluted to the final percentage of sepiolite into the extruder in order to obtain the corresponding sheets for characterization. The use of these modifiers reduces the sepiolite–sepiolite interactions, which favors a better dispersion, and alignment of nanofibers that translate in effectiveness in mechanical properties [7].

Drying conditions for the master were 80 °C for 7 h, and 120 °C for 7 h for the pristine PET. The drier used was a CRAMER-TROCKNER model PK 100/300F. With the aim of simplification, from now on virgin PET will be referred to as PET.

The industrial extruder used was a Luigi Bandera SpA twin screw extruder, from LINPAC Packaging. In this extruder it was obtained the nanocomposite’s sheet that then is taken to a KIEFEL GmbH thermoforming machine to obtain the desired final trays. The trays chosen for this project are MAP trays, with the following dimensions: 18 cm width, 25 cm length, and 45 mm depth. In this work it will be referred to as B1825-45 tray. The nanocomposite trays are sent to a poultry packer (Sada, Nutreco. Spain). There, 2 kg of breast chicken are packed in a modified atmosphere containing 70% CO_2_, 20% O_2_, and 10% N_2_ in each tray. Control trays are packed in the same way, in a PET tray. Then, samples are taken for microbiological analysis of mesophilic aerobes and *Enterobacteriaceae* for 14 days. In this work, the microbiological quality of chicken fillets was assessed by determining the number of mesophilic aerobic bacteria, and *Enterobacteriaceae*. These analyses will help us determine if microbial load of those species in chicken, packed in nanocomposite trays, is lower than that packed in regular PET, in order to have an idea of food sanitary quality.

### 3.1. TGA

Thermogravimetric Analysis (TGA) was used to determine nanosepiolite percentage within the nanocomposite sheets. The analyses were performed in a Mettler Toledo 851e equipment, using a procedure in two steps:

First step: from 50 to 600 °C at 20 °C/min under nitrogen atmosphere.

Second step: from 600 to 900 °C at 20 °C/min under air atmosphere.

### 3.2. Permeability

The permeability analyses were done on sheet samples; specimens were taken from the extruded sheets before going to thermoforming into trays.

Oxygen transmission rate (OTR) was measured in an OXTRAN with a volumetric sensor (Oxtran SS 2/20, MOCON. Barcelona, Spain). Previously to the analysis the samples were conditioned, 48 h under an atmosphere with 0% RH. Oxygen transmission rate was measured at 23 °C and 0% RH following Standard ASTM D3985-17 and the effective area exposed to permeation was 50 cm^2^.

### 3.3. Puncture Test

Plastic products are more prone to fail when submitted to a multiaxial impact, rather than to a slow-motion load. In many applications, packaging materials are exposed to penetrating damages, which lead to barrier properties and package integrity deterioration. Thus, it is very important to obtain packaging materials with good impact strength properties that help preserve the food until its use. 

These impact tests were done in an MTS-831 equipment, following ISO 6603-2:2000 methodology [8]. The speed used was 4.4 m/s and tests were at room temperature (23 °C).

The specimen, with an effective diameter of 40 mm, is hold with two anchor rings, then the impactor (φ_impactor_ = 20 mm) hits on the specimen center from below. The curve strength versus strain is registered for each sample, together with absorbed energy (E). However, it is very important to describe the failure mode in order to know if the material is going to break in a fragile way, a ductile, or in any of the intermediate modes in between [8]. In a ductile break (D), the specimen breaks slowly deforming the material with the absorbed E, while the additional, non-absorbed E, is used to extend the crease (Dc). On the contrary, in a fragile break (F) the crease is spread quickly, suddenly, and totally, causing the break of the sample.

### 3.4. Compression Test

Lateral compression tests and stiffness were tested on a Hounsfield H1KS Benchtop equipment following LINPAC Packaging internal procedures for trays. For each trial, between 65 and 70 trays were tested.

### 3.5. Microbiological Tests

Twelve specimens of the nanocomposite trays were analyzed for each material, and sixteen for control.

For total viable count (TVC) determination, 25 g of superficial meat are taken aseptically. Samples are mixed with 225 mL of buffered peptone water and is then homogenized in a Stomacher^®^ (dilution 1:10). After that, 1 mL sample is taken from the main dilution and then the dilutions needed to obtain an appropriate number of microorganisms are done. The incubation time and temperature for mesophilic aerobes are 72 h at 30 °C and a Petrifilm Aerobic Count Plate is used (ISO 4833-1, 2003 [9]), and 24 h at 37 °C for Enterobateriacae using a 3M Petrilm to help counting (ISO 21528-2, 2004 [10]). Bacterial count results are expressed in log_10_ of colony-forming units per gram of meat (log cfu/g).

Microbiological analyses were done, on 3 samples per day and treatment, on the following days post packaging: 2, 7, 10, and 14. At the same time, head space gases were measured to see the evolution during the microbial study, using an OXIBABY-M O2/CO2 (WITT-GASETECHNIK. Witten, Germany). The specimens were kept at 5 °C during all that period.

## 4. Results and Discussion

It has been analyzed before [6], the effect caused by different nanosepiolite masterbatches concentration (one with 20%, and the other one with 10%) on the final nanocomposite properties. It was concluded that those nanocomposites coming from a less concentrated master had better homogeneity, as well as viscosity and mechanical properties of the final material. Thus, for this study, a 10% nanosepiolite master was aimed. This percentage of sepiolite is theoretical because after dosing the sepiolite in an industrial extruder, the final amount of nanoclay in the nanocomposite changes. This is due to the difficulty of adding a powder to an industrial extruder at 600 kg/h. In this way, a PET masterbatch with 7.66% sepiolite modified with MEMO (MB1), and another one with 8.56% of sepiolite modified with AMEO (MB2) were used in this study determined by TGA.

Table 1 shows the percentages of sepiolite in the nanocomposites, after dilution of the masters into the PET industrial extruder.

### 4.1. Permeability

The permeability was analyzed on industrially extruded sheet samples with different thickness. Permeability to oxygen, calculated as OTR, improved in all nanocomposite’s samples compared to that of pure PET. The improvement can be up to 30% with 1.37% of nanosepiolite. It is shown the OTR/Sheet Thickness, which is calculated dividing the permeability, given in OTR units, by the sample thickness. It was observed that increasing the clay concentration did not change the permeability of the samples substantially (Figure 1a). In addition, as stated by Ke and Yongping [11] and tested in this work, the processability of the nanocomposite is much more difficult when increasing the amount of nanosepiolite over 3% [6,12]. It is possible to decrease the amount of nanoclay in the samples, as long as the nanoparticles are well dispersed and oriented within the matrix [13,14,15,16]. However, if the sepiolite content is too low it will not do the job and if it is too much, PET matrix viscosity will decrease and it will not disperse properly, opening the path for gas and vapor molecules.

Comparing the nanocomposite samples to a PET-EVOH-PE laminated sheet, which is the sheet normally used when high barrier is required, we can see 20% improvement in the sample containing 1.37% of nanosepiolite modified with MEMO (sample M2). There are no mayor differences in terms of permeability performance between the two silanes used in this study.

Looking at Figure 1b it can be seen that, in order to achieve the same permeability of a 450 µm PET sample, a nanoPET composites could decreased its thickness in about 100–150 µm. These results are really important in terms of reducing PET consumption, which is good for both company savings in materials and transport, and for the environment.

### 4.2. Mechanical Properties of Nanocomposite Trays

In order to overcome its transport from factory to houses, including customer’s handling, trays must be tough enough. That is why it is so important to test the packages with tests that can simulate its treatment once at the market.

These tests are done on the industrially extruded sheet before the thermoforming process. Registered curves for impact tests show a maximum which is related to the initial damage on the sheet, corresponding to the starting point of the fissure that will develop in a fracture. Analyzing the curves obtained in this test, maximum load and its associated deformation can be known, as well as perforation energy. On the other hand, this test shows the way the sample breaks allowing us to define the failure mode of each specimen.

Thickness is measure on extruded sheet, and the results shown in Table 2 are the average obtained in all the samples width (795 mm). Since this measure is of great importance in impact results, we will compare samples with the same thickness. Thus, for a sheet with 465 µm, a sample M5 with 1.65% nS shows an 8% improvement in impact strength [14]; nevertheless, the impact energy decreases 18% in the nanocomposite compared to that of pure PET (with 467 µm) (Table 2). This means there is less energy for deformation, but it takes a little higher force to break the nanocomposite sheet. The same behavior is seen in nanocomposites with MEMO, but impact and energy values are a little lower than in those modified with AMEO.

A PET sheet (100% virgin) shows a ductile behavior, whilst nanocomposite materials evidence a fragile component when using MEMO modified sepiolite (see Table 2). However, samples modified with AMEO do not show fragile behavior but a ductile with a forming crease one. Otherwise, there are no notable differences in nanocomposites impact behavior depending on the organomodifier used.

Regarding sample’s thickness, when increasing this value, maximum load, impact energy, and ductility at break increases in all the samples no matter the sepiolite content or its organic modifier.

In order to normalize result to thickness, data in Newtons were divided by the thickness of each sample. The results show that samples modified with AMEO improve its maximum load at impact when increasing nanosepiolite content, whereas samples with MEMO have poorer results. Impact load increases in all samples with AMEO over that of pure PET, with a 19% increase in sample containing 1.88% nanosepiolite (M6). Meanwhile, the maximum load improvement in MEMO samples is of 9% over that of virgin PET.

### 4.3. Compression Tests

Compression tests results are done with the tray in vertical position since is, generally, the most critical force MAP trays are going to be subjected to. Results indicate that the nanocomposite trays have more resistance to compression forces compare to the pure PET trays.

Nanocomposites are more resistant to lateral compression than pure PET. The improvement can be 66.7% for the sample M6 containing 1.88% nS.

In the following graph (Figure 2) it is shown the stiffness per gram. The results show that for the same tray weight the stiffness increase can be up to 85% with 1.88% nanosepiolite AMEO modified. Thus, the addition of nanosepiolite to a PET matrix results in increased stiffness and decreased ductility, as seen in these results [6,15]. This could be due to a good interaction between the PET matrix and the sepiolite. The high aspect ratio of this clay is key in this interaction, together with the organomodifiers used [16,17]. It is also seen here an increase in stiffness with the amount of sepiolite when using AMEO as organomodifier, an increase of 0.23 g nanosepiolite produces an improvement of 40% in stiffness. Whereas in a sample with nearly double sepiolite content modified with MEMO (1.09 g more), there is an increase of just 4%.

It is seen that nanocomposites sheets with around 1.6% nanosepiolite need 150 µm less thickness in order to generate the same stiffness as a pure PET samples. Thus, a sample with 1.65% nanosepiolite modified with AMEO (M5) has the same stiffness than a sample that weights 8 g more in pure PET.

As it is shown in this paper, results with both AMEO and MEMO as organic modifiers for the sepiolite have quite similar performance. However, those nanocomposites with AMEO are more resistant to lateral compression forces, a little stronger and less brittle. Moreover, between the two nanocomposite’s types there are some other differences which are important for good industrial extrusion and thermoforming processes. These differences are mainly that AMEO nanosepiolite seems to disperse better into the PET matrix [6], and its processability in an industrial extruder was a little better, showing a more stable behavior.

As a function of the results obtained, the sheet chosen for the next step is that containing between 1.5 and 1.9% of nanosepiolite modified with 2% of AMEO. At the same time, it will be used the trays with intermediate thickness. Thus, the sheet sample chosen to thermoformed trays to be used in the next step is M5. These trays were thermoformed in order to do microbiological analysis and migration test studies.

### 4.4. Microbial Analysis in Packed Chicken Breast

Microbiological analysis is done on breast chicken packed in the same trays as discussed previously, a B1825-45 tray made with M5 sheet (Tray A1, 465 m) and using a virgin PET tray in 601 µm as control (Tray A0). This thickness is the regular one that would be used for this type of product. The tray thickness refers to the average thickness of the sheet before thermoforming (the average value of all sheet width).

Mesophilic aerobes and *Enterobacteriaceae* family include not only pathogenic species, but environmental species as well, which often appear in the food manufacturing environment without posing any health hazard. In fresh food, high number counts are not recommended, although an elevated count does not imply the presence of pathogenic flora. However, the total count reflects sanitary quality of the analyzed products. If the total count is high, testing of specific pathogens can be done. At the same time, a low count does not mean the sample is pathogen free, it depends on the composition of the microbiota [18]. This control should be done in accordance with Commission Regulation (EC) No 2073/2005.

In the following graphs (Figure 3) it is shown microbiology parameter’s growth, together with atmosphere gases evolution with packing time for both samples chosen; which is to say, 2 kg of breast chicken packed in trays A1, and also the control samples which are packed in pure PET (A0). In both cases, the top film is BOPET coated with AlOx. Results shown in the graphics correspond to the average count obtain for the three samples examined, for each period (2, 7, 10, and 14 days).

Generally, in hygienically handled fresh food, with a limited shelf life, if the count of microorganism is high it will decrease the serviceable time of the products. The chicken microbiological quality was initially good in both trays, being at day 2 (first count analyses) lower than 4 log (cfu/g) [19,20]. At day fourteenth, none of the samples had arrived at 7 log (cfu/g) which is the maximum limit stablished for fresh chicken [21,22,23]. Looking at mesophilic aerobes results (Figure 3a,d), it can be seen the values of the two trays (A0, A1) are quite similar up until the fourteenth day, where results on nanocomposite’s tray show lower numbers of colony forming units of mesophilic aerobes in those trays made with nanocomposite, up to one order of magnitude lower (from log (cfu/g) 6.57 to 5.25).

Regarding total count of *Enterobacteriaceae* (EB), the average value of PET trays analyzed is 3.83 log (cfu/g) (8.25 × 10^3^ ufc/g) whilst this number is 2.98 log (cfu/g) (9.67 × 10^2^ ufc/g) for the nanocomposite tray. Thus, nanosepiolite trays have lower microbiological charge. EB remains practically constant and very low from day 2 to 14 for the nanocomposite trays, starting at an average log (cfu/g) of 2.83 and ending, at 14 days, at 2.98, whereas in PET trays, although the starting count of *Enterobacteriaceae* is lower, log (cfu/g) 2.43, at day 14 it is of 3.80. The explanation for the lower growth of bacterial counts could be due not only to CO_2_ [24] but also to N_2_ capability of inhibiting bacterial growth [25,26].

Figure 3b,d shows head space gases evolution for the breast chicken packed in nanocomposite’s trays and in PET tray. CO_2_ content decrease is mainly due to product absorption into the food [21,22,23,24,25,26,27,28], but also to loss of the gas through permeation across the plastic. Major CO_2_ absorption takes place during the first two days in the two samples, resulting in an equilibrium concentration, lower than the initial [29]. The absorption of CO_2_ depends on partial pressure in the head space of the packed tray [30] and also on the product itself (for example, chicken with skin or without it). In this study, these factors are the same for the two trays mentioned. However, the final CO_2_ content is approximately 20% higher in nanocomposite’s trays. Thus, in spite of the lower thickness of the new trays, the decrease in permeability helps in keeping CO_2_ inside the package.

Regarding O_2_, this was maintained quite constant throughout the 14 days due to initial amount being very similar to atmosphere O_2_ concentration (approximately 20.9%). O_2_ percentage increased during the first two days, due to the high decrease of CO_2_ in that period, which caused loss of volume, and to the lack of diffusion of O_2_ to the exterior since its partial pressure is nearly the same in and outside the package. After those days, O_2_ remained quite constant for the reason already mention. It can be also seen in the gas’s evolution figures that N_2_ expands at the same time that CO_2_ is absorbed into the chicken.

The improved barrier performance of the PET/sepiolite nanocomposites has, therefore, produced a decreased on microbial counts in the poultry trays due to their ability to keep modified atmosphere gases longer [12,31,32].

### 4.5. Migration

Global migration test for food contact plastics are done in a variety of food simulants, depending on the food type it is aimed to contain (fatty food, vegetables, etc.) and the storage conditions. In this work, global migration tests have been done following specifications under UNE-EN 1186-14 using iso-octane as a fatty food simulant.

For these analyses, the samples chosen were M3 and M6 (PET + 1.78% nS_MEMO and PET + PET + 1.88% nS_AMEO, respectively), since these are the ones with the highest sepiolite content. Global migration limits within European Regulation (UE) No 10/2011 and RD 866/2008 are 10 mg/dm^2^ or 60 mg/kg. Results on both trays are lower than 1.0 mg/dm^2^; thus, both trays (M3 and M6) comply with global migration limits for the chosen simulant and under the test conditions.

Regarding specific migration limits for MEMO and AMEO (0.05 mg/dm^2^), being the results 0.04 mg/dm^2^ for both MEMO and AMEO, it can be concluded that both trays also comply with the limits stablished in Directive 2002/72/EC and its subsequent amendments (Regulation EC 975/2009) for this product types. Actual legislation for food contact materials does not contain specific migration limits for natural silicates (except asbestos) such as sepiolite. However, since sepiolite is a magnesium silicate, the global migration of silicon and magnesium in iso-octane has been analyzed in order to know if there is any sepiolite migration from tray material to the packaged food. Results for silicon migration are lower than 0.05 mg/kg in both trays (M3 and M6), and results for magnesium are lower than 1.0 mg/kg. For this reason, it is concluded that migration of these two elements to the packaged food is well under the legislation limit.

## 5. Conclusions

The results obtained in the study have showed that the global properties of the trays have been improved.

It is possible to improve PET permeability by 30% with a nanocomposite containing just 1.37% of sepiolite in its matrix. It would be possible to decrease tray thickness in about 100–150 µm in order to obtain the same permeability of a 450 µm PET sample. These results would imply company savings in terms of materials and transport, and also it would be great for the environment, using less raw materials.

Mechanical properties have been also measured. In the impact test, it has been seen an improvement in the maximum load at impact in all the samples but in M3, which value is equal to that of the PET. In those samples modified with AMEO, impact energy is the same as to a PET sample with 1.88% sepiolite, whilst for the MEMO modified samples the best one, nearly equal to that of PET in impact energy is that one with 1.2% clay.

Failure behavior is ductile for a 100% virgin PET and has a fragile component in the nanocomposite sheet with MEMO modified sepiolite. However, samples modified with AMEO does not show fragile behavior but a ductile with a forming crease one. Otherwise, MEMO and AMEO samples show similar mechanical performances.

Nanocomposites trays are stiffer than PET ones and show better performance to lateral compression forces. This type of compression is most critical for sealed trays, since the top film tends to produce tension and, if the tray is not stiff enough, it will bend or even collapse easily. This would give a poor packaging impression to customers. It has been seen that it is possible to lighten trays with the use of nanosepiolite clay, which is very important both industrially and environmentally.

The final nanocomposite trays show lower numbers of colony forming units of mesophilic aerobes in those trays made with nanocomposite, up to one order of magnitude lower (from log(cfu/g) 6.57 to 5.25). EB count remains practically constant and very stable from day 2 to 14 for the nanocomposite trays, being the final count one order of magnitude lower than that obtained in a virgin PET tray (from log(cfu/g) 3.83 to 2.98).

It is important to note that nanocomposite trays are 136 µm less thick than that of the PET serving as control. Thus, lighter nanosepiolite trays with lower microbiological charge can be produced at industrial machines. Moreover, these nanocomposite trays comply with European Regulation (UE) No 10/2011 and RD 866/2008 for materials intended to be in contact with food products.

## Figures and Tables

**Figure 1 foods-10-01188-f001:**
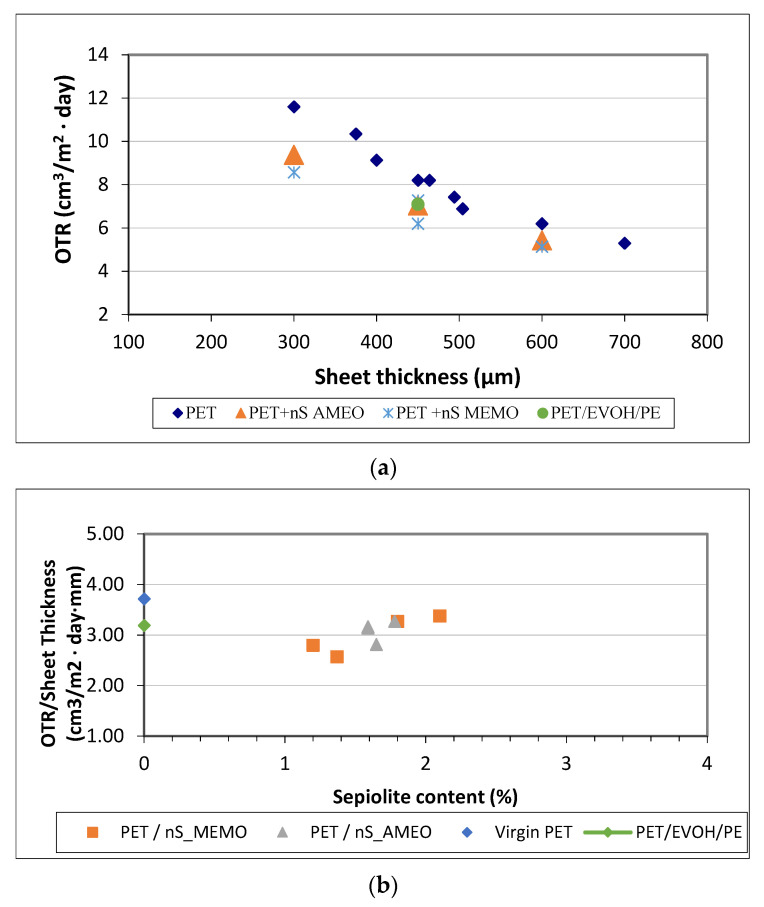
Oxigen transmission rate (OTR)/Sheet Thickness as a function of sepiolite content on the nanocomposite samples (**a**); OTR for nanocomposites as a function of sheet thickness (**b**).

**Figure 2 foods-10-01188-f002:**
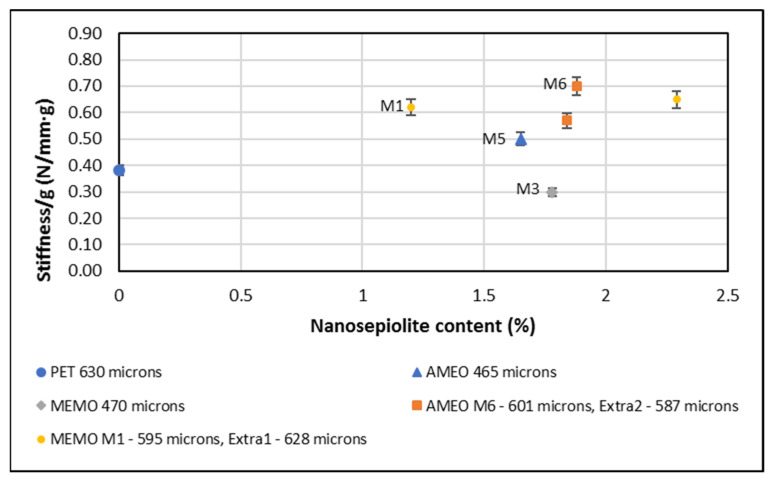
Nanocomposites stiffness per gram compared to that of pure PET.

**Figure 3 foods-10-01188-f003:**
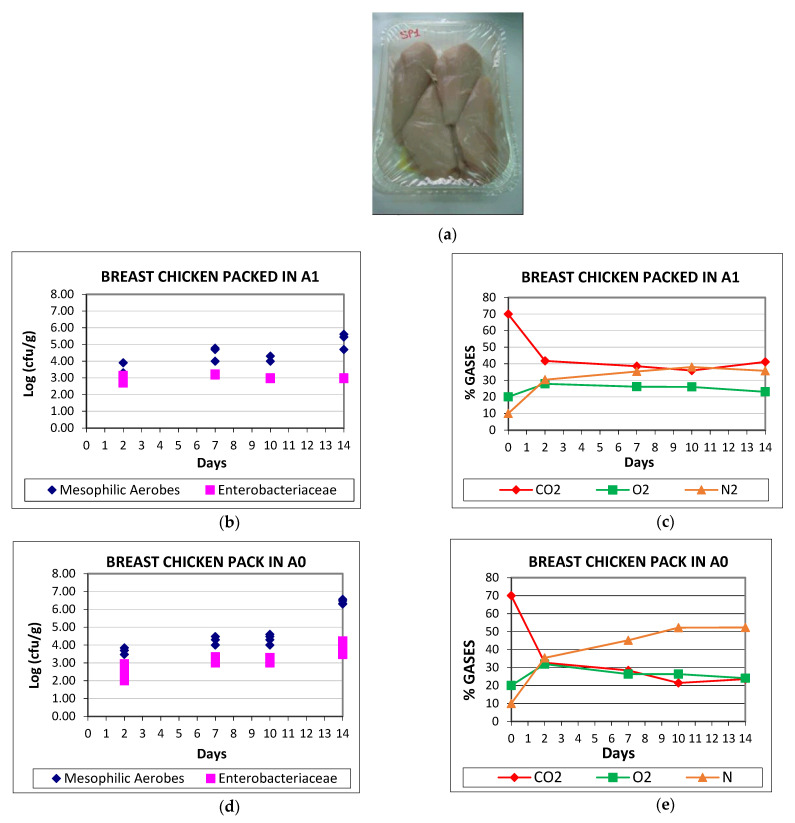
Picture of the packed chicken (**a**), aerobic mesophilic and *Enterobacteriaceae* count (**b**,**d**) and gases concentration evolution in 2 kg chicken breast (**c**,**e**), packed in A1 and A0 nanocomposite tray, respectively.

**Table 1 foods-10-01188-t001:** Samples and sepiolite content on the nanocomposites.

Master	Sample	Nanocomposite	TGA (% nS)
-	M0	Virgin PET	0
MB1	M1	PET + 1.20% nS_MEMO	1.20
	M2	PET + 1.37% nS_MEMO	1.37
	M3	PET + 1.78% nS_MEMO	1.78
MB2	M4	PET + 1.58% nS_ AMEO	1.59
	M5	PET + 1.65% nS_ AMEO	1.65
	M6	PET + 1.88% nS_ AMEO	1.88

**Table 2 foods-10-01188-t002:** Impact puncture results for nanocomposites containing nanosepiolite modified with MEMO (3-metracyloxypropil trimetoxysilane) and with AMEO (3-aminopropyltriethoxysilane).

Sample	TGA (% nS)	Thickness (µm)	Max Load (N)	Puncture Energy (J)	Failure Mode *	Picture
**MEMO**						
M0	0	467 ± 2	720 ± 16	6.6 ± 0.8	D	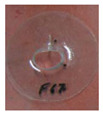
M2	1.37	351 ± 12	560 ± 42	2.6 ± 0.6	FD	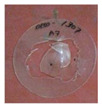
M3	1.78	470 ± 11	730 ± 13	5.0 ± 2.4	FD-D	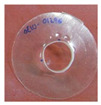
M1	1.20	595 ± 15	1000 ± 23	8.3 ± 0.9	D	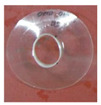
**AMEO**						
M0	0	467 ± 2	720 ± 16	6.6 ± 0.8	D	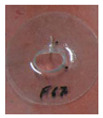
M4	1.59	320 ± 6	520 ± 15	3.1 ± 0.7	FD-Dc	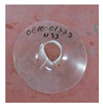
M5	1.65	465 ± 9	780 ± 34	5.4 ± 0.7	Dc	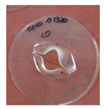
M6	1.88	601 ± 12	1100 ± 40	8.5 ± 0.4	D	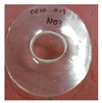

* Failure Modes: D—Ductile, FD—transition Fragile/Ductile, Dc—Ductile with crease.

## Data Availability

The data presented in this study are available on request from the corresponding author. The data are not publicly available due to privacy restrictions of the companies involved.

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
