# Peer review of "Application of PET/Sepiolite Nanocomposite Trays to Improve Food Quality"

_foods, 2021, doi:10.3390/foods10061188_

Round 1
Reviewer 1 Report
I find your article interesting, but some small changes are required.

Author Response
The answer is in the submitted file
Reviewer 2 Report
Manuscript 1294344 reports on the mechanical and physicο-chemical properties of PET/Sepiolite Nanocomposites including the shelf life of chicken meat packaged in this material. This study, in my opinion, has a lot more of Material Science than Food Science into it. Furthermore, even though the authors talk about shelf life, they never ran such a study (including physicochemical-pH, TVB-N, TMA, TBA analysis, and sensory evaluation). Regarding microbiological analysis they only monitored TVC and Enterobacteriaceae. Chicken spoils mainly through the growth of the psychrotrophic pseudomonads, Brochothrix thermosphacta (spoilage specific organism), Enterobacteriaceae and in case of foods stored under MAP, by lactic acid bacteria. Of these microbial groups, only TVC and Enterobacteriaceae were monitored. Also, when shelf life of chicken is investigated, Salmonella spp. should always be determined.
Another basic problem of the study is that during the assessment of the experimental packaging materials, only the tray component was evaluated. No data is provided on the properties of the top sealing film (BOPP) i.e. the permeability of this film is critical regarding product shelf life. Obviously, the package is composed of top web plus base tray.
A third general comment is the need for substantial improvement in the use of English language. The authors should consult with a native English speaker.
My detailed comments follow the text sequence:
l.24-26: no one can talk about shelf life if one does not run sensory evaluation on the product vs. storage time. Also, the authors cannot refer to food poisoning and nutritional/sensorial value of a food without determining these aspects.
l.36: abbreviations should follow definition of each term
l.39-40: polymers are equally used for the packaging of plant and animal products
l.41 and 43: terms such as ‘enlarge shelf life’ and ‘viability of foods’ are not used in Food Science
l.52: foods ‘do not travel’ , they are ‘distributed’
l.57: poultry is not ‘manipulated’, it is ‘handled’
l.60: change ‘amount’ to ‘mixture’
l.75: delete: ’ At the end of this paper’
l.79: the nomenclature part should precede the text
l.82-83: the statement ‘PET-EVOH-PE laminated sheet, is a PET sheet laminated with
EVA/PA/EVOH/PA/PE flexible film’ is totally wrong
l.89: do the authors mean ‘stated’ by ‘stayed’ ?
l.113: provide information on company city and country
l.118: how was this gas mixture selected ? According to the literature ( www.dansensor.com) mixtures: 70/30, O2/CO2 or 20/80 O2/CO2 or 3070 CO2/N2 are used for raw poultry MA Packaging
l.121-122: see my general comment on the determination of poultry shelf life
l.136: change ‘upgraded’ to ‘conditioned’
l.141: rewrite sentence in proper English
l.143: properties do not ‘decrease’. They ‘deteriorate’
l.145: we refer other to ‘equipment’ or ‘instrument’ What is an ‘instrumental equipment’ ?
l.164: Food scientists do not use the term ‘microorganism count’. They use the term Total viable count (TVC) or Aerobic plate count (APC)
l.168: incubation conditions for TVC are usually ‘370C/48 h’ rather than ‘300C/72h’
l.136 and 175: include city and country of company
l.175-176: section on statistical analysis is missing
l.177: rewrite sentence in proper English
l.189-190: rewrite sentence in proper English
l.200: rewrite sentence in proper English
l.222: change ‘manipulation too ‘handling’
l.241: rewrite sentence in proper English
Fig.1b: I question the permeability values reported for PET sheet. A 12 μm thick PET film has a permeability of ~140 ml O2/m2/day/atm. This should correspond to 2.8 ml O2/m2/day/atm for a 300 μsheet of PET instead of 12 shown in Fig. 1b
l.262: change ‘submitted’ to ‘subjected’
l.298: the authors cannot refer to product shelf life just by determining TVC and Enterobacteriaceae ! See my general comment
Fig3 c,e: It is difficult to believe data in these Figures. As aerobic spoilers grow they consume oxygen within the package. This is not shown in Fig. 3 for oxygen
Fig. 4 and l.429: had the authors expected such an extension of product shelf life they should have run the experiment for 30-35 days. Such an attempted extrapolation is not valid when we refer to such an important parameter as product shelf life
l.359: the authors mean ‘decrease’ and not ‘increase’
Based on all the above, I recommend rejection of the manuscript
Author Response
The answer is in the submitted file

Reviewer 3 Report
Dear authors,
The results give statements about the use of PET/Sepiolite Nanocomposites for and Shelf Life of Chicken .
General comment:
- The article should be revised regarding spelling errors and sentence structure
- Line 71: Quantitative statement about aspect ratio is missing (is important to understand the barrier improvement factor)
- Line 80-92: Experimental design: The relationship of the laminates described in chapter 2 and the composition of the laminate of the later used packages is not clear. Please add this information, eg. with a figure that explains the experimental setup
- Figure 1: There must be a mistake in Figure 1: The unit of a permeation coefficient is quantity of material * thickness / (area * time * gradient); eg cm³*mm/(m²*d*bar)! The permartion coefficient of PET/EVOH/PE is typically below that of PET. Therefore please revise Figure 1 and the conclusions related to Figure 1
- Line 345 ff and Line 428 f: Extrapolation of shelf life on basis of the existing data is not possible and maybe even risky, (for example) because you do not consider pathogens. I strongly recommend to skip these sections
- Line 359: Do you mean decrease instead of increase?
Best reagards
km
Author Response
The answer is in the submitted file

Round 2
Reviewer 2 Report
REVIEWER 2
Manuscript 1294344 reports on the mechanical and physicο-chemical properties of PET/Sepiolite Nanocomposites including the shelf life of chicken meat packaged in this material. This study, in my opinion, has a lot more of Material Science than Food Science into it. Furthermore, even though the authors talk about shelf life, they never ran such a study (including physicochemical-pH, TVB-N, TMA, TBA analysis, and sensory evaluation). Regarding microbiological analysis they only monitored TVC and Enterobacteriaceae. Chicken spoils mainly through the growth of the psychrotrophic pseudomonads, Brochothrix thermosphacta (spoilage specific organism), Enterobacteriaceae and in case of foods stored under MAP, by lactic acid bacteria. Of these microbial groups, only TVC and Enterobacteriaceae were monitored. Also, when shelf life of chicken is investigated, Salmonella spp. should always be determined. We understand the reviewer’s comment and the paper has been modified, avoiding the use of the term “shelf life”, with the aim of correcting this point.
Reviewer’s new comment
It seems that the authors have not understood the significance of my comment on ‘shelf life’ since in the title, abstract and elsewhere of revised text they still use the term ‘shelf life which they never determined. Also microbiological analysis of chicken meat is incomplete (see my initial comment above)
Another basic problem of the study is that during the assessment of the experimental packaging materials, only the tray component was evaluated. No data is provided on the properties of the top sealing film (BOPP) i.e. the permeability of this film is critical regarding product shelf life. Obviously, the package is composed of top web plus base tray. The top film has been kept apart because it was not a parameter that would change. The top film has been always the same during the analyses done, and we wanted to focus on the tray composition effect.
Reviewer's new comment
This is not a satisfactory answer. The package consists of two separate parts (top web and tray) both contributing to the preservation of chicken meat. However no real information is provided on the top web.
l.82-83: the statement ‘PET-EVOH-PE laminated sheet, is a PET sheet laminated with EVA/PA/EVOH/PA/PE flexible film’ is totally wrong. Sorry, I´m not sure I understand this. The PET-EVOH-PE sheet used in this project is been produced at LINPAC facilities. Its production is done laminating an EVA/PA/EVOH/PA/PE film onto a PET sheet at the same extrusion machine (which has a lamination unit).
Reviewer's new comment
What I meant was that the sheet PET-EVOH-PE is different from EVA/PA/EVOH/PA/PE
l.118: how was this gas mixture selected ? According to the literature ( www.dansensor.com) mixtures: 70/30, O2/CO2 or 20/80 O2/CO2 or 3070 CO2/N2 are used for raw poultry MA Packaging. The gas mixture has been chosen for the poultry company. Another reviewer has suggested to delete that sentence.
Reviewer's new comment
The answer is not satisfactory. The authors should know which gas mixtures are recommended for the packaging of chicken under MAP
l.121-122: see my general comment on the determination of poultry shelf life. The term “shelf life” has been substituted by “microbial load”.
Reviewer's new comment
‘Shelf life’ and ‘microbial load’ are two different things. What does ‘microbial load’ mean ? Does it mean TVC or something else ?
l.175-176: section on statistical analysis is missing. It could be interesting for future works but, unfortunately, it was not consider for this one.
Reviewer's new comment
Whatever data is presented should be the result of statistical analysis. This was not done in the revised text
l.298: the authors cannot refer to product shelf life just by determining TVC and Enterobacteriaceae ! See my general comment. The term “shelf life” has been substituted by “Microbial analysis”.
Reviewer's new comment
As I have previously indicated, microbiological analysis carried out was incomplete
Fig3 c,e: It is difficult to believe data in these Figures. As aerobic spoilers grow they consume oxygen within the package. This is not shown in Fig. 3 for oxygen. As stayed by previous authors, the increase in oxygen could be due “the permeability differences
of the gases through the packaging material and the dissolution of CO2 in the aqueous phase of the chicken muscle” (Reference 21). The diffusion rate for CO2 is generally 20 times greater than O2. In our results it is seen an increased during the two first days, and then the oxygen concentration starts to decrease.
Reviewer's new comment
Oxygen permeability of a high barrier material such as that used in the present study is very low. Thus, the assumption made by the authors does not hold.
Author Response
The answer to this reviewer is in the attached file

Reviewer 3 Report
In the revised version, the authors followed the comments of my review.
Author Response
The language has been reviewed